# The association between exposure to interferon-beta during pregnancy and birth measurements in offspring of women with multiple sclerosis

**Sarah Burkill**[ID][1,2]*, **Pia Vattulainen**[3], **Yvonne Geissbuehler**[4], **Meritxell Sabido Espin**[5], **Catrinel Popescu**[6], **Kiliana Suzart-Woischnik**[7], **Jan Hillert**[8], **Miia Artama**[9], **Auli Verkkoniemi-Ahola**[10], **Kjell-Morten Myhr**[11], **Sven Cnattingius**[2], **Pasi Korhonen**[3], **Scott Montgomery**[2,12,13], **Shahram Bahmanyar**[1,2]

1 Centre for Pharmacoepidemiology, Karolinska Institutet, Stockholm, Sweden, 2 Division of Clinical Epidemiology, Department of Medicine Solna, Karolinska Institutet, Stockholm, Sweden, 3 EPID research, Helsinki, Finland, 4 Novartis Pharma AG, Basel, Switzerland, 5 Merck group, Darmstadt, Germany, 6 Biogen, Maidenhead, United Kingdom, 7 Bayer AG, Berlin, Germany, 8 Department of Clinical Neuroscience, Karolinska Institutet, Stockholm, Sweden, 9 National Institute for Health and Welfare, Helsinki, Finland, 10 Department of Neurology, Helsinki University, Helsinki, Finland, 11 Department of Clinical Medicine, University of Bergen, Bergen, Norway, 12 Clinical Epidemiology and Biostatistics School of Medical Sciences, Örebro University, Örebro, Sweden, 13 Department of Epidemiology and Public Health, University College London, London, United Kingdom

* sarah.burkill@ki.se

## Abstract

### Background

Interferon-beta (IFN-beta) is a commonly used treatment for multiple sclerosis (MS). Current guidelines recommend cessation of treatment during pregnancy, however the results of past studies on the safety of prenatal exposure to IFN-beta have been conflicting. A large scale study of a population of MS women is therefore warranted.

### Objectives

To assess whether, among those born to women with MS, infants prenatally exposed to IFN-beta show evidence of smaller size at birth relative to infants which were not prenatally exposed to any MS disease modifying drugs.

### Methods

Swedish and Finnish register data was used. Births to women with MS in Sweden and Finland between 2005–2014 for which a birth measurement for weight, height, and head circumference was available were included. The exposure window was from 6 months prior to LMP to the end of pregnancy.

**Data Availability Statement:** The data that support the findings of this study are available from the Swedish National Board of Health and Welfare

(Prescribed Drug Register, National Patient Register, and Swedish Medical Birth Register), Karolinska Institute (MS register), and Statistics Sweden (Total Population Register). In Finland, the respective data are available from the National Institute of Welfare and Health (Care Register for Health Care, and Medical Birth Register, Register) and from the Social Insurance Institute (National Reimbursement Register and National Prescription Register). Restrictions apply to the availability of these data, which were used under license for this study. The data supporting these findings can be obtained by applying to the respective data holders in Sweden and Finland. For Sweden, data access queries can be sent to Registerservice@socialstyrelsen.se for the PDR, NPR, and MBR. For the MS register, data access queries can be sent to the register co-ordinator, anna.cunningham@ki.se. For the total population register, data access queries can be sent to scb@scb.se. For Finland, data access queries can be sent to info@tfl.fi for the MBR and CRHC. For the NRR and NPR, data access queries can be sent to tilastot@kela.fi.

**Funding:** This study was funded by Bayer AG, Biogen Netherlands B.V., Merck KGaA, and Novartis Europharm Limited. YG is employed by Novartis Pharma AG, MS is employed by Merck group, CP is employed by Biogen and KSW is employed by Bayer AG. Novartis Pharma AG, Merck group, Biogen and Bayer AG provided support in the form of salaries for authors YG, MS, CP, and KSW. The funders were not involved in data collection or analysis, decision to publish, or preparation of the manuscript beyond the contributions of these authors, however they were able to comment on study design in the initial stages of the project. The specific roles of the authors employed by the funders are articulated in the 'author contributions' section. PV and PK are employed by EPID research. EPID research provided support in the form of salaries for authors PV and PK, and were involved in data collection and decision to publish, but were not involved in data analysis or manuscript preparation beyond the contributions of these authors. The specific roles of the authors are articulated in the 'author contributions' section.

**Competing interests:** I have read the journal's policy and the authors of this manuscript have the following competing interests: This study was funded by Bayer AG, Biogen Netherlands B.V., Merck KGaA, and Novartis Europharm Limited. SB reports no personal conflict of interest, but works for CPE which receives funding from third parties including pharmaceutical companies. PV reports

## Results

In Sweden, 411 pregnancies were identified as exposed to IFN-beta during the exposure window, and 835 pregnancies were counted as unexposed to any MS DMD. The corresponding numbers for Finland were 232 and 331 respectively. Infants prenatally exposed to interferon-beta were on average 28 grams heavier (p = 0.17), 0.01 cm longer (p = 0.95), and had head circumferences 0.14 cm larger (p = 0.13) in Sweden. In Finland, infants were 50 grams lighter (p = 0.27), 0.02 cm shorter (p = 0.92) and had head circumferences 0.22 cm smaller (p = 0.15) relative to those unexposed.

## Conclusions

This study provides evidence that exposure to IFN-beta during pregnancy does not influence birth weight, length, or head circumference.

## Introduction

Multiple sclerosis (MS) is an immune mediated disease causing demyelination and axonal loss in the central nervous system (CNS) [1]. Women with MS often cease treatment during pregnancy, in part due to immunological adaptations during pregnancy including suppression of T-cell activity[2], which occur to allow foetal growth without the foetus being recognised as a foreign agent by the immune system, often reducing symptoms[3]. Interferon beta (IFN-beta) is a commonly used disease modifying treatment for MS, with current guidelines recommending cessation during pregnancy. Pregnant women are often excluded from clinical trials, meaning the effects of prenatal exposure are not known, resulting in a preference to cease treatment. If the pregnancy is planned, decisions need to be made as to whether treatment should be continued. However not all pregnancies are planned and some women will conceive whilst on treatment, making foetal exposure to IFN-beta inevitable in some instances. The possible effects of this are not known with certainty. Results from smaller scale studies generally show no adverse pregnancy outcomes for mothers exposed to IFN-beta during pregnanc[4–8]. However, some studies have concluded that exposure could potentially be detrimental, making the benefit–risk assessment for mother and child and therefore the decision to continue or discontinue treatment after conception difficult[9]. Given conflicting outcomes in past research [4–9], an investigation into potential adverse pregnancy outcomes related to foetal exposure to IFN-beta using larger populations of women with MS rather than smaller samples is therefore warranted.

This study utilized administrative data from Sweden and Finland to compare birth measurements (birth weight, height, and head circumference) of infants of women with MS exposed to IFN-beta, and infants of women with MS unexposed to MS disease modifying drugs (DMD).

## Methods

### Study population and data sources

Pregnancies resulting in a live birth to women with a diagnosis of MS were identified using the Medical Birth Registers between the years of 2005 and 2014 in Sweden and Finland. These dates were selected because the PDR in Sweden began in 2005, so it would not be possible to

no personal conflict of interest, but works for EPID Research which is a contract research organization and thus its employees have been and currently are working in collaboration with several pharmaceutical companies. YG is an employee of Novartis Pharma AG. MS is an employee of Merck KGaA. CP is an employee of Biogen. KSW is an employee of Bayer AG. JH has received honoraria for serving on advisory boards or steering committees for Biogen, Merck, Sanofi-Genzyme and Novartis and speaker's fees from Biogen, Novartis, Merck-Serono, Bayer-Schering, Teva and Sanofi-Genzyme. He has served as P.I. for projects, or received unrestricted research support from, BiogenIdec, Merck-Serono, TEVA, Sanofi-Genzyme and Bayer-Schering. MA reports no conflict of interest. AM reports no conflict of interest. KMM has received unrestricted research grants to his institution and/or scientific advisory board or speakers honoraria from Almirall, Biogen, Genzyme, Merck, Novartis, Roche and Teva; and has participated in clinical trials organized by Biogen, Merck, Novartis, and Roche. PK works for EPID Research which is a contract research organization and thus its employees have been and currently are working in collaboration with several pharmaceutical companies. SM has received funding from AstraZeneca, and funding for MS related research from F. Hoffman-La Roche AG, and Novartis International AG. ShB reports no personal conflict of interest, but works for CPE which receives funding from third parties including pharmaceutical companies. There are no patents, products in development or marketed products to declare. This does not alter our adherence to all the PLOS ONE policies on sharing data and materials.

identify prescriptions of IFN-beta prior to this date. The corresponding date was chosen as the beginning of the study in Finland to ensure results were comparable. The end of the study was 31st December 2014, because at the time of the data application this was the most recently available data. In Sweden, the mother's diagnosis of MS was identified using the National Patient Registers (NPR) (International Classification of Diseases tenth revision [ICD 10] code G35), and the National Multiple Sclerosis Register (MSR). Informed consent is required for a patient to be included in the MSR. Data for identification of exposure to IFN-beta and other disease modifying treatments was retrieved from the Prescribed Drugs Register (PDR), which records all collected prescriptions within Sweden and allows for identification of treatments through the use of anatomical therapeutic chemical classification (ATC) system codes. The MS Register was also used to identify when treatment had been initiated in hospital. In Finland, an MS diagnosis was identified using the National Reimbursement Register, and the Care Register for Health Care, and exposure to IFN-beta was retrieved from the Finnish PDR, using ATC codes.

Only observations for which a birth height, birth weight, and birth head circumference measurement had been recorded were included. All three measurements needed to be present for the birth to be included in the study to ensure the same infants were being compared throughout the analyses. All pregnancies to women with MS within Sweden and Finland between 2005 and 2014 were included in this population based study.

## Exposure and outcome

Exposure to IFN-beta was counted as having occurred if the individual collected a prescription of IFN-beta during the exposure window, here defined as up to 6 months prior to the last menstrual period (LMP) until the end of the pregnancy according to the PDR of Sweden or Finland, or if a date of initiation for IFN-beta was recorded in the MSR in Sweden. It was assumed that the woman would have been exposed to IFN-beta in the 3 months before LMP or later, because the prescriptions are intended to last 3 months. Exposure to any MS DMD was also identified using the same registers and exposure window (see S1 Table for the list of ATC codes) aside from cladribine and mitoxantrone, for which exposure 6 months prior (purchase 9 months prior) to LMP was considered as exposed.

Within the MSR, exposure definitions are reliant on dates of treatment initiation and cessation. The accuracy of using such dates to determine exposure may be considered questionable. Clinicians who only temporarily cease treatment due to the pregnancy would not necessarily consider this a cessation of treatment. In these instances, date of cessation indicating the treatment was stopped during pregnancy would go unrecorded. It is also possible that no cessation date was recorded because the woman continued treatment during the exposure window. To examine whether results changed when applying different exposure criteria from the MSR, a sensitivity analysis identified all pregnancies with recorded treatment before the 6 months prior to LMP exposure window but with no cessation date during the exposure window. This analysis provides numbers exposed and mean values for each outcome, rather than comprising the main definition due to the previously mentioned limitations. It was only possible to conduct this sensitivity analysis using Swedish data, since such information is not available in the Finnish registers.

Pregnancies exposed to IFN-beta only were compared to pregnancies unexposed to any MS DMD. Birth measurements for weight in grams, height in cm, and head circumference in cm were used in a continuous form, and information was taken directly from the Medical Birth Registers of Sweden and Finland.

## Statistical analysis

Linear regression using generalized estimating equations (GEEs) was undertaken in which the mother's ID was used as a cluster identifier which demarcates siblings. Maternal age at LMP, gestational age in weeks, maternal smoking status, and within Sweden highest maternal educational attainment (separated into compulsory school or less, upper secondary, or higher education) were considered to be possible confounders, and included in the adjusted models. Sex of the newborn was also included in the model due to its association with birth measurements. An analysis studying differently exposed siblings was also undertaken and included the same confounders as covariates. Adjusting for maternal age at LMP meant birth order was considered in this model.

A sensitivity analysis which compared women exposed to any MS DMD (including but not limited to IFN-beta) to women unexposed to any MS DMD was also undertaken to assess whether the outcomes differed to when IFN-beta exposure only was studied.

Continuous measures which compare mean measurements between groups are the primary outcomes used in this paper.

## Ethics statement

In Finland, the study was given a positive opinion by the Helsinki University Hospital Ethics Committee (Finland; 159/13/03/00/2016). Data permit approvals were granted by the National Institute for Health and Welfare (Dnro THL/635/5.05.00/2016) and the Social Insurance Institution (Dnro 42/522/2016). In Sweden, the study was approved by the Regional Ethical Review Board in Stockholm (Sweden; 2016/874-31/2). Data permit approvals were granted by National Board of Health and Welfare (Dnr 23981/2016) and Swedish MS Registry (Dnr 53). All individuals within Sweden and Finland are automatically included in administrative records through entry into national registers at birth or immigration. Informed consent is not gained for this. These databases are primarily administrative in nature, and are not set up with research as the intended outcome, however they are commonly used for such purposes. The datasets are held at the institutions listed in the data access statement, and were fully anonymised by the holding institutions before delivery.

## Results

In Sweden, 411 pregnancies were identified as exposed to IFN-beta during the exposure window for which all birth measurements were available, and 835 pregnancies were counted as unexposed to any MS DMD. The corresponding numbers for Finland were 232 and 331, respectively. Within Sweden, there were 1131, and within Finland 442 women with MS registered as having a pregnancy during the study time frame. In Sweden, there were 101 pregnancies comprised of 50 sibling sets identified as being differently exposed to IFN-beta. The corresponding numbers for Finland were 83 pregnancies comprised of 41 sibling sets. The study population characteristics showed most women in Sweden had received higher education (studying at university) as their highest educational attainment (data on education were unavailable in Finland), and had a parity of 2 overall. When considering all pregnancies, the unexposed cohort had on average a slightly older age at LMP relative to the exposed cohort in Sweden, and in Finland the ages were very similar at LMP (Table 1). Gestational age at birth was not significantly different among the exposed and unexposed cohorts, with mean gestational age of 39.7 weeks for the exposed and 39.5 weeks (p = 0.10) for the unexposed cohorts in Sweden, and 39.4 weeks for the exposed and 39.5 weeks (p = 0.67) for the unexposed cohorts in Finland. Births prior to 22 weeks are not recorded in the medical birth registers. Very early

**Table 1. Exposed and unexposed cohort characteristics.**

| | Sweden | | | |
|---|---|---|---|---|
| | All | | Differently exposed siblings | |
| | Exposed | Unexposed | Exposed | Unexposed |
| **Number of pregnancies** | 411 | 835 | 50 | 51 |
| **Mean (SE) Gestational age, weeks,** | 39.7 (0.1) | 39.5 (0.1) | 40.0 (0.2) | 39.2 (0.3) |
| **Mean (SE) Birth weight, grams** | 3465.9 (27.7) | 3414.8 (19.4) | 3475.5 (66.3) | 3346.6 (81.7) |
| **Mean (SE) Birth height, cm's** | 50.1 (0.1) | 50.0 (0.1) | 50.3 (0.4) | 49.7 (0.4) |
| **Mean (SE) Head circumference, cm** | 35.0 (0.1) | 35.0 (0.1) | 35.0 (0.2) | 34.7 (0.3) |
| **Infant Sex** | | | | |
| **Male (%)** | 207 (50.4) | 441 (52.8) | 24 (48.0) | 26 (51.0) |
| **Female (%)** | 204 (49.6) | 394 (47.2) | 25 (52.0) | 25 (49.0) |
| **Mean (SE) maternal age, years** | 31.3 (0.2) | 32.3 (0.2) | 31.0 (0.6) | 30.9 (0.5) |
| **Maternal education** | | | | |
| **Compulsory school or less (%)** | 20 (4.9) | 56 (6.7) | 2 (4.0) | 2 (3.9) |
| **Upper secondary (%)** | 113 (27.5) | 242 (30.0) | 15 (30.0) | 17 (33.3) |
| **Higher education (%)** | 277 (67.4) | 534 (64.0) | 33 (66.0) | 32 (62.8) |
| **Missing data (%)** | 1 (0.2) | 3 (0.4) | 0 (0) | 0 (0) |
| **Smoking status** | | | | |
| **Smoker (%)** | 18 (4.4) | 54 (6.5) | 45 (90.0) | 47 (92.2) |
| **Nonsmoker (%)** | 378 (92.0) | 740 (88.6) | 3 (6.0) | 0 (0) |
| **Not known (%)** | 15 (3.7) | 41 (4.9) | 2 (4.0) | 4 (7.8) |
| | Finland | | | |
| | All | | Differently exposed siblings | |
| | Exposed | Unexposed | Exposed | Unexposed |
| **Number of pregnancies** | 232 | 331 | 41 | 42 |
| **Mean (SE) Gestational age, weeks,** | 39.4 (2.4) | 39.5 (1.9) | 39.4 (2.9) | 40.0 (1.2) |
| **Mean (SE) Birth weight, grams** | 3357.5 (628.3) | 3410.4 (541.0) | 3306.6 (649.2) | 3508.4 (441.7) |
| **Mean (SE) Birth height, cm's** | 49.5 (3.1) | 49.6 (2.5) | 49.2 (3.8) | 49.9 (1.9) |
| **Mean (SE) Head circumference, cm** | 34.5 (2.2) | 34.8 (1.7) | 34.4 (2.6) | 35.0 (1.4) |
| **Infant Sex** | | | | |
| **Male** | 117 (50.4) | 171 (51.7) | 18 (43.9) | 18 (43.9) |
| **Female** | 115 (49.6) | 160 (48.3) | 23 (56.1) | 24 (57.1) |
| **Mean (SE) maternal age, years** | 30.0 (4.2) | 30.6 (4.5) | 30.0 (4.2) | 30.6 (4.5) |
| **Maternal education** | Not available | Not available | Not available | Not available |
| **Smoking status** | | | | |
| **Smoker (%)** | 33 (14.2) | 49 (14.8) | 2 (4.9) | 3 (7.1) |
| **Nonsmoker (%)** | 195 (84.1) | 277 (83.7) | 39 (95.1) | 39 (92.9) |
| **Not known (%)** | 4 (1.7) | 5 (1.5) | 0 (0) | 0 (0) |

births under 32 weeks are rare. S1–S3 Figs show the distribution of birth measurements and allow for identification of outliers.

In the adjusted analyses, infants prenatally exposed to IFN-beta were on average 28 grams heavier (p = 0.17) in Sweden, and 50 grams lighter (p = 0.26) in Finland than those unexposed (Table 2). For birth height, those exposed to IFN-beta were 0.01 cm longer (p = 0.95) in Sweden, and 0.02 cm shorter (p = 0.92) in Finland compared to those unexposed. For head circumference, those in Sweden had measurements 0.14 cm larger (p = 0.13) and in Finland 0.22 cm smaller (p = 0.15) relative to those unexposed (unadjusted analyses shown in Table 2, adjusted analyses shown in Table 3).

**Table 2. GEE's unadjusted- differences in mean weight, height, and head circumference between exposed and unexposed cohorts.**

| | Unadjusted | | | | | |
| --- | --- | --- | --- | --- | --- | --- |
| | Weight | P-value | Height | P-value | Head circumference | P-value |
| **Sweden** | | | | | | |
| **Overall** | 51.1 (37.4) | 0.17 | 0.12 (0.2) | 0.48 | 0.20 (0.1) | 0.08 |
| **Differently exposed siblings** | 128.8 (110.0) | 0.24 | 0.53 (0.5) | 0.27 | 0.31 (0.3) | 0.34 |
| **Finland** | | | | | | |
| **Overall** | -52.8 (59.6) | 0.38 | -0.10 (0.3) | 0.74 | -0.26 (0.2) | 0.21 |
| **Differently exposed siblings** | -198.2 (126.7) | 0.12 | -0.71 (0.8) | 0.36 | -0.57 (0.5) | 0.3 |

Analysis of groups of siblings in which at least one sibling was exposed, and at least one sibling was unexposed was undertaken (see Tables 2 and 3). Overall, differences between siblings exposed and siblings unexposed to IFN-beta during pregnancy were minimal, and not statistically significant. The same was true when considering exposure to any MS DMD, relative to non-exposure to MS DMD (see S3–S5 Tables).

## Discussion

Infants born to women with MS exposed to IFN-beta during pregnancy did not show evidence of decreased intrauterine growth relative to infants born to MS women unexposed to IFN-beta during pregnancy. There were also no differences in mean gestational age at birth. No differences were found when comparing infants prenatally exposed to any MS DMD, relative to infants not exposed to MS DMD. This confirms evidence from some past studies which report no adverse effects for infants prenatally exposed to IFN-beta[5, 7, 10], and refutes the findings of other studies which have found IFN-beta is associated with birth measurement[11].

Birth measurements which are substantially below average have been reported to be associated with a number of adverse health outcomes, including behavioural difficulties such as ADHD[12], and other health conditions including coronary heart disease and diabetes[13, 14], although the extent to which these effects are confounded by social or genetic factors is incompletely understood[15]. The increased risks of adverse outcomes later in life highlights the importance of studying such measures.

Gestational age is one of the most critical factors determining birth measurements[16], with expected foetal weight gain of 24–26 grams per day in the third trimester for low risk pregnancies[17]. Gestational ages at birth were comparable across exposure groups, and maternal ages at LMP, indicating exposure to IFN-beta does not result in earlier gestational

**Table 3. GEE's adjusted- differences in mean weight, height, and head circumference between exposed and unexposed cohorts.**

| | Adjusted* | | | | | |
| --- | --- | --- | --- | --- | --- | --- |
| | Weight | P-value | Height | P-value | Head circumference | P-value |
| **Sweden** | | | | | | |
| **Overall** | 27.8 (20.1) | 0.34 | 0.01 (0.1) | 0.95 | 0.14 (0.1) | 0.13 |
| **Differently exposed siblings** | -21.6 (77.1) | 0.78 | -0.10 (0.4) | 0.78 | -0.05 (0.3) | 0.85 |
| **Finland** | | | | | | |
| **Overall** | -50.3 (45.1) | 0.27 | -0.02 (0.2) | 0.92 | -0.21 (0.2) | 0.15 |
| **Differently exposed siblings** | -83.6 (79.8) | 0.30 | 0.07 (0.4) | 0.85 | -0.008 (0.3) | 0.98 |

*Adjusted for gestational age, sex of the newborn, smoking status of the mother, and maternal age at LMP

age at birth, and will therefore not have an impact on birth measurements through this mechanism.

One explanation as to why IFN-beta may not be influencing birth measurements or gestational age could be due to its pharmacokinetic characteristics. The placental barrier is a semi-permeable tissue which separates foetal and maternal blood, and is only permeable for substances with a low molecular weight (between 600 and 800 Dalton)[18]. IFN-beta is categorized as a polypeptide with a molecular weight of 22.kDa (kiloDalton) for IFN-beta 1a and 18.5kDa for IFN-beta 1b[19], which is too large to permeate the placental barrier. The likelihood of IFN-beta therefore being able to directly affect the development of the foetus through permeation of foetal blood is unlikely, and suggests the lack of an effect of IFN-beta on foetal growth measurements is biologically plausible.

The maternal identification numbers included in our data allowed for observation of specific sets of siblings. This was particularly useful, as it enabled us to consider the effect of exposure to IFN-beta on an individual, relative to their unexposed sibling or siblings. In contrast to regular population analysis, sibling analysis controls by design for unmeasured shared familial (genetic and environmental) factors. Differences between the sibling exposure groups for all included birth measurements were not statistically significant for either Sweden or Finland.

Younger women are more likely to be of lower parity, and to have smaller mean birth measurements for their offspring[20]. Previous studies have indicated the effect of young maternal age on low birth height and weight measurements does not persist when behavioural, biological, and socioeconomic confounders have been controlled for[21, 22], indicating that maternal age may primarily be a confounder. Our study indicated the age at LMP for the pregnancies exposed to IFN-beta was on average lower than the age at LMP for the pregnancies unexposed to any MS DMD. This has the potential to influence birth measurements through potentially unmeasured confounders[23], even though maternal age at LMP itself has been included in the adjusted models.

A strength of this register study was our ability to include all women with MS who had a pregnancy that ended with a birth in both Sweden and Finland during the study period. Previous studies have relied on samples which can be self-selecting, and have reduced power due to smaller numbers enrolled than is possible using register data[24]. Inclusion of a population of pregnant women, with information available for all pregnancies to a particular patient group, removes the possibility of selection bias through for example over-recruitment of low- or high-risk pregnancies, improving the reliability of the results.

Potential limitations of the study should also be considered. It was not possible to know whether exposure identified using the Prescribed Drugs Register resulted in an exposed pregnancy, because we cannot be certain the treatment was taken by the mother. Only data which stated the drug had been dispensed was available for prescriptions, with the assumption then made that the treatment was taken as instructed. Previous studies into different treatments have indicated that pregnant women are less likely to adhere to treatment than other patient groups[25, 26], which increases the likelihood that dispensation does not necessarily equate to exposure. Only pregnancies for which information was recorded for all three measurements (weight, height, and head circumference at birth) were included in the study. If missingness is differential by exposure status, bias may be induced. We were limited in the variables we were able to adjust for, due to limited information being provided in registers. For example, potentially informative data on diet and physical activity are not available. A live birth had to have occurred for inclusion in the study population. The Medical Birth Registers of Sweden and Finland record births occurring at 22 weeks or later of pregnancy. Elective terminations are additionally not identifiable in the Swedish data. If the rates of elective termination and spontaneous abortion differ by exposure group, there is the potential for bias to be induced.

In summary, the evidence from this large population based study indicate no association between IFN-beta exposure and fetal growth or gestational age among infants of women with MS.

## Supporting information

**S1 Fig. Birth weight in grams by gestational age, according to interferon-beta exposure status.**
(DOCX)

**S2 Fig. Birth height in cms by gestational age, according to interferon-beta exposure status.**
(DOCX)

**S3 Fig. Head circumference by gestational age, according to interferon-beta exposure status.**
(DOCX)

**S1 Table. ATC codes and brand names used to identify interferon-beta exposure.**
(DOCX)

**S2 Table. Exposed to interferon-beta according to all possibly exposed sensitivity analysis (Sweden only).**
(DOCX)

**S3 Table. Exposure to any MSDMD's sensitivity analysis.**
(DOCX)

**S4 Table. GEE OLS unadjusted models.**
(DOCX)

**S5 Table. GEE OLS adjusted models.**
(DOCX)

**S6 Table. Exposure to any MSDMD's all possibly exposed sensitivity analysis (Sweden only).**
(DOCX)

## Acknowledgments

We would like to acknowledge the European Interferon Beta Pregnancy Study Group for their contribution to this manuscript

## Author Contributions

**Conceptualization:** Yvonne Geissbuehler, Meritxell Sabido Espin, Catrinel Popescu, Kiliana Suzart-Woischnik, Jan Hillert, Miia Artama, Auli Verkkoniemi-Ahola, Kjell-Morten Myhr, Sven Cnattingius, Pasi Korhonen, Scott Montgomery, Shahram Bahmanyar.

**Data curation:** Sarah Burkill, Pia Vattulainen.

**Formal analysis:** Sarah Burkill, Pia Vattulainen.

**Methodology:** Sarah Burkill, Pia Vattulainen.

**Project administration:** Sarah Burkill.

**Resources:** Yvonne Geissbuehler, Meritxell Sabido Espin, Catrinel Popescu, Kiliana Suzart-Woischnik, Jan Hillert, Miia Artama, Auli Verkkoniemi-Ahola, Kjell-Morten Myhr, Sven Cnattingius, Pasi Korhonen, Scott Montgomery, Shahram Bahmanyar.

**Supervision:** Scott Montgomery, Shahram Bahmanyar.

**Writing – original draft:** Sarah Burkill.

**Writing – review & editing:** Sarah Burkill, Pia Vattulainen, Yvonne Geissbuehler, Meritxell Sabido Espin, Catrinel Popescu, Kiliana Suzart-Woischnik, Jan Hillert, Miia Artama, Auli Verkkoniemi-Ahola, Kjell-Morten Myhr, Sven Cnattingius, Pasi Korhonen, Scott Montgomery, Shahram Bahmanyar.

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
