## [Decision Letter · Decision Letter 0]

30 Sep 2019

PONE-D-19-21273

The association between exposure to interferon-beta during pregnancy and birth measurements in offspring of women with multiple sclerosis

PLOS ONE

Dear Miss Burkill,

Thank you for submitting your manuscript to PLOS ONE. After careful consideration, we feel that it has merit but does not fully meet PLOS ONE’s publication criteria as it currently stands. Therefore, we invite you to submit a revised version of the manuscript that addresses the points raised during the review process.

Two expert individuals in the field have reviewed the manuscript and found the work to be of interest. Further clarification is needed on the study design, power, and why select dates were chosen. Revisions are also required in the Tables.

We would appreciate receiving your revised manuscript by Nov 14 2019 11:59PM. To enhance the reproducibility of your results, we recommend that if applicable you deposit your laboratory protocols in protocols.io, where a protocol can be assigned its own identifier (DOI) such that it can be cited independently in the future. For instructions see: http://journals.plos.org/plosone/s/submission-guidelines#loc-laboratory-protocols

We look forward to receiving your revised manuscript.

Kind regards,

Cheryl S. Rosenfeld, DVM, PhD

Academic Editor

PLOS ONE

Journal Requirements:

2. In the ethics statement in the manuscript and in the online submission form, please provide additional information about the patient records used in your retrospective study. Specifically, please ensure that you have discussed whether all data were fully anonymized before you accessed them and/or whether the IRB or ethics committee waived the requirement for informed consent. If patients provided informed written consent to have data from their medical records used in research, please include this information.

4. Thank you for stating the following in the Financial Disclosure section:

'This study was funded by Bayer AG, Biogen Netherlands B.V., Merck KGaA, and

Novartis Europharm Limited. The study sponsors could comment on the study design,

the interpretation of data, and the writing of the manuscript. The study sponsors had no

role in the data collection or analysis, or the decision to submit the paper for

publication.'.   

We note that one or more of the authors are employed by a commercial company: Novartis Pharma AG, Merck group, Biogen & Bayer AG.

5. We note you have included a table to which you do not refer in the text of your manuscript. Please ensure that you refer to Table 3 in your text; if accepted, production will need this reference to link the reader to the Table.

Additional Editor Comments (if provided):

Reviewers' comments:

Reviewer's Responses to Questions

**Comments to the Author**

1. Is the manuscript technically sound, and do the data support the conclusions?

Reviewer #1: Yes

Reviewer #2: Yes

2. Has the statistical analysis been performed appropriately and rigorously? 

Reviewer #1: I Don't Know

Reviewer #2: Yes

3. Have the authors made all data underlying the findings in their manuscript fully available?

Reviewer #1: No

Reviewer #2: Yes

4. Is the manuscript presented in an intelligible fashion and written in standard English?

Reviewer #1: Yes

Reviewer #2: Yes

5. Review Comments to the Author

Reviewer #1: This is a registry study, with limitations inherent to that methodology, of women with MS in Sweden and, separately in Finland, who were exposed to BIFN and to other DMDS compared to MS women who delivered live births who were not.

1. Reviewer would like to see in Methods or elsewhere comments by the authors on the powering of the study. Although no significant difference in bodily measures of newborns were uncovered in either Sweden or Finland, there were trends in each country’s cohort. Can a statement be made regarding what degree of difference would these sized groups be empowered to detect?

2. This study examined the size of newborns. To be counted, a live birth must have occurred. One limitation not mentioned is termination of pregnancies (miscarriages or abortions) which may have occurred in these women. This should be mentioned. Is there any way to uncover the rate of selective or non-selective terminations and how that may have affected the study results?

3. The investigators identified sibling pairs of mothers with MS, in which one was exposed and the sibling was not exposed to BIFN, and found no differences in body measures of infant siblings based on exposures. It would also be of interest to know what the power for detecting a difference would be. For example, would it be powered sufficiently to detect a 10% difference in one of the measures? And, could the authors mention what degree of difference in head circumference or other measures is considered significant?

4. The study results are averages. Is there a way to identify the numbers of outliers in each group? Numbers/rates of premature infants?

5. The authors speculate that, due to size, beta-interferons will not cross the placental barrier due to its low permeability for high molecular weight substances. Are there not data (even from animal studies) to cite which can provide more definitive information on whether beta-interferons can cross the placenta, or not?

Reviewer #2: The following points are addressed to the authors: A)Why was the period 2005-2014 chosen? B) Please note the number of sibs studied in the man text. C) On page 7, reference 11 (Betaseron Pregnancy registry) did NOT report birth measurement abnormalities with IFN beta exposure. D)Tables 2 and 3 are totally confusing and not truly discussed in the text. Why are sibs in there? Where is Sweden?

6. PLOS authors have the option to publish the peer review history of their article (what does this mean?). If published, this will include your full peer review and any attached files.

Reviewer #1: No

Reviewer #2: No

---

## [Author Response · Author response to Decision Letter 0]

30 Oct 2019

We would like to thank the reviewers for their constructive comments, which have been invaluable for the improvement of the manuscript. Our responses to specific comments can be seen below. 

Reviewer #1:

This is a registry study, with limitations inherent to that methodology, of women with MS in Sweden and, separately in Finland, who were exposed to BIFN and to other DMDS compared to MS women who delivered live births who were not.

Comment 1: Reviewer would like to see in Methods or elsewhere comments by the authors on the powering of the study. Although no significant difference in bodily measures of newborns were uncovered in either Sweden or Finland, there were trends in each country’s cohort. Can a statement be made regarding what degree of difference would these sized groups be empowered to detect?

Response 1: Power calculations are an attempt to assess how precisely estimates reflect the ‘true’ value of the population based on sample values and variance. However, the study population was not sampled. Instead we included the entire population of women with MS with a live birth. In the method section, we stress that the entire population of available individuals was included in the study (page 4 paragraph 2).

“All pregnancies to women with MS within Sweden and Finland between 2005 and 2014 were included in this population based study”.

Comment 2: This study examined the size of newborns. To be counted, a live birth must have occurred. One limitation not mentioned is termination of pregnancies (miscarriages or abortions) which may have occurred in these women. This should be mentioned. Is there any way to uncover the rate of selective or non-selective terminations and how that may have affected the study results?

Response 2: This has now been added to the limitation section on page 10 paragraph 1. It is not possible to identify elective terminations for the study dates within Sweden, because they are not recorded in the birth register or any other patient registers. The birth register only includes births over 22 weeks, making miscarriage difficult to identify. This has now been discussed in the limitations section. 

“A live birth had to have occurred for inclusion in the study population. The Medical Birth Registers of Sweden and Finland record births occurring at 22 weeks or later of pregnancy. Elective terminations are additionally not identifiable in the Swedish data. If the rates of elective termination and spontaneous abortion differ by exposure group, there is the potential for bias to be induced.”

Comment 3: The investigators identified sibling pairs of mothers with MS, in which one was exposed and the sibling was not exposed to BIFN, and found no differences in body measures of infant siblings based on exposures. It would also be of interest to know what the power for detecting a difference would be. For example, would it be powered sufficiently to detect a 10% difference in one of the measures? And, could the authors mention what degree of difference in head circumference or other measures is considered significant?

Response 3: It is not possible to state exactly what difference would be counted as statistically significant, and what non-significant, because this also depends in large part on variance and the extent to which results are heterogenous. It is not possible with real data showing non-significance to state that a difference of X would result in an outcome being statistically significant. It may simply be that there are no differences between the cohorts. A lack of statistical power is one possible reason for non-significance, but we cannot prove either way whether it is the case. 

Comment 4: The study results are averages. Is there a way to identify the numbers of outliers in each group? Numbers/rates of premature infants?

Response 4: Gestational ages were comparable across the two cohorts, and preterm delivery was not statistically significantly more common in either group relative to the other. This was included the results section on page 7 paragraph 1. There were very few outliers in which births were very early. Differences between cohorts would therefore be difficult to detect. In order to show distributions of birth measurements more clearly, scatterplots which showed birth measurement by gestational age have been included in the supplementary materials.

“Gestational age at birth was not statistically significantly different between the exposed and unexposed cohorts, with mean gestational age of 39.7 weeks for the exposed and 39.5 weeks (p=0.10) for the unexposed cohorts in Sweden, and 39.4 weeks for the exposed and 39.5 weeks (p=0.67) for the unexposed cohorts in Finland. Births prior to 22 weeks are not recorded in the Medical Birth Registers. Very early births under 32 weeks are rare Figures 1-3 in the supplementary materials show the distribution of birth measurements and allow for identification of outliers.”.

Comment 5: The authors speculate that, due to size, beta-interferons will not cross the placental barrier due to its low permeability for high molecular weight substances. Are there not data (even from animal studies) to cite which can provide more definitive information on whether beta-interferons can cross the placenta, or not? 

Response 5: The literature, as far as we are aware, is based on observations (even in animal studies) for exposed vs unexposed subjects. These references are included currently in the manuscript (references 18 and 19). We have not been able to identify studies which take e.g. specific placental measurements of IFN-beta levels. If the reviewer can provide any we are happy to add them.

Reviewer #2: 

The following points are addressed to the authors:

Comment 1: Why was the period 2005-2014 chosen? 

Response 1: The Prescribed Drugs Register started in 2005 in Sweden. Therefore, it would not be possible to identify instances of IFN-beta prescription dispensation prior to this date. To ensure comparability of results, we decided to use the same study period in both Sweden and Finland. The date 2014 was chosen as study end, because at the point of data order, this was the most recent available data. This has been clarified in the methods section in page 4 paragraph 1.

“These dates were selected because the PDR in Sweden began in 2005, so it would not be possible to identify prescriptions of IFN-beta prior to this date. The corresponding date was chosen as the beginning of the study in Finland to ensure results were comparable. The end of the study was 31st December 2014, because at the time of the data application this was the most recently available data.”

Comment 2: Please note the number of sibs studied in the man text.

Response 2: For the differently exposed sibling analysis, the number of sibling groups has now been included in the results section on page 7 paragraph 1. Only pregnancies to women with MS were included in this study. This means that any children born before the woman was diagnosed, are not included in the study population, so sibling sets will only refer to births after MS diagnosis. It therefore seemed to make more sense to simply report the number of women included in the study. This has now also been added into the results (page 7 paragraph 1).

“Within Sweden, there were 1131, and within Finland 422 women with MS registered as having a pregnancy during the study time frame. In Sweden, there were 101 pregnancies comprised of 50 sibling sets identified as being differently exposed to IFN-beta. The corresponding numbers for Finland were 83 pregnancies comprising 41 sibling sets. ”

Comment 3: On page 7, reference 11 (Betaseron Pregnancy registry) did NOT report birth measurement abnormalities with IFN beta exposure. 

Response 3: This reference has been moved and added into the list of references demonstrating no measurement abnormalities.

Comment 4: Tables 2 and 3 are totally confusing and not truly discussed in the text. Why are sibs in there? Where is Sweden?

Response 4: Table 2 is the unadjusted, and table 3 the adjusted results of the generalised estimating equations. The first row in tables 2 and 3 should be labelled to show those results pertain to Sweden. We thank the referee for pointing this out. This has now been rectified. It is not all siblings, but differently exposed siblings which are included there (where at least one sibling is prenatally exposed and at least one sibling is prenatally unexposed to IFN beta) because differently exposed siblings were compared as part of the analysis. The results of table 3 (the adjusted analysis) are covered in the results section on page 7 paragraph 2.

“In the adjusted analyses (table 3), infants prenatally exposed to IFN-beta were on average 28 grams heavier (p=0.17) in Sweden, and 50 grams lighter (p=0.26) in Finland than those unexposed. For birth height, those exposed to IFN-beta were 0.01 cm longer (p=0.95) in Sweden, and 0.02 cm shorter (p=0.92) in Finland compared to those unexposed. For head circumference, those in Sweden had measurements 0.14 cm larger (p=0.13) and in Finland 0.22 cm smaller (p=0.15) relative to those unexposed (unadjusted analysis shown in table 2, adjusted analysis shown in table 3).

---

## [Decision Letter · Decision Letter 1]

13 Dec 2019

The association between exposure to interferon-beta during pregnancy and birth measurements in offspring of women with multiple sclerosis

PONE-D-19-21273R1

Dear Dr. Burkill,

We are pleased to inform you that your manuscript has been judged scientifically suitable for publication and will be formally accepted for publication once it complies with all outstanding technical requirements.

With kind regards,

Cheryl S. Rosenfeld, DVM, PhD

Section Editor

PLOS ONE

Additional Editor Comments (optional):

Reviewers' comments:

Reviewer's Responses to Questions

**Comments to the Author**

1. If the authors have adequately addressed your comments raised in a previous round of review and you feel that this manuscript is now acceptable for publication, you may indicate that here to bypass the “Comments to the Author” section, enter your conflict of interest statement in the “Confidential to Editor” section, and submit your "Accept" recommendation.

Reviewer #2: (No Response)

2. Is the manuscript technically sound, and do the data support the conclusions?

Reviewer #2: Yes

3. Has the statistical analysis been performed appropriately and rigorously? 

Reviewer #2: Yes

4. Have the authors made all data underlying the findings in their manuscript fully available?

Reviewer #2: Yes

5. Is the manuscript presented in an intelligible fashion and written in standard English?

Reviewer #2: Yes

6. Review Comments to the Author

Reviewer #2: The authors have answered all points in a satisfactory manner. I might just add in the Abstract/Results: No significant differences were noted between interferon beta exposed vs. unexposed infants.

7. PLOS authors have the option to publish the peer review history of their article (what does this mean?). If published, this will include your full peer review and any attached files.

Reviewer #2: Yes: Patricia K. Coyle, MD

---

## [Editor Report · Acceptance letter]

19 Dec 2019

PONE-D-19-21273R1 

The association between exposure to interferon-beta during pregnancy and birth measurements in offspring of women with multiple sclerosis 

Dear Dr. Burkill:

I am pleased to inform you that your manuscript has been deemed suitable for publication in PLOS ONE. Congratulations! Your manuscript is now with our production department. 

With kind regards,

on behalf of

Dr. Cheryl S. Rosenfeld 

Section Editor

PLOS ONE